# Seismic Imaging of the Mesozoic Bedrock Relief and Geological Structure under Quaternary Sediment Cover: The Bolmin Syncline (SW Holy Cross Mountains, Poland)

**Bartosz Owoc [1,*]**, **Artur Marciniak [1]**, **Jan Dzierżek [2]**, **Sebastian Kowalczyk [2]** and **Mariusz Majdański [1]**

[1]  Institute of Geophysics, Polish Academy of Sciences, 01-452 Warsaw, Poland; amarciniak@igf.edu.pl (A.M.); mmajd@igf.edu.pl (M.M.)
[2]  Faculty of Geology, University of Warsaw, 02-089 Warsaw, Poland; j.dzierzek@uw.edu.pl (J.D.); s.kowalczyk@uw.edu.pl (S.K.)
*  Correspondence: bowoc@igf.edu.pl

**Abstract:** The clear and detailed images of geological structures that can be obtained by seismic methods are one of the main drivers of their popularity in geological research. The quality of final geophysical images and models relies strongly on the amount of data that goes into them. Analysing several complementary seismic datasets allow an improved interpretation. Responding to this challenge, this article proposed an optimal combination of geophysical methods for near-surface applications. Multi-channel analysis of surface waves, first-arrival travel-time tomography, and ground-penetrating radar were the key supports for standard reflection seismic imaging. Ease of use and fast and cheap acquisition are some of the advantages of the mentioned methods. Considering that all recorded wave fields required minimal additional processing while offering a significant improvement in the final stack, it was worth the extra effort. Thanks to that, the better-estimated velocity filed allowed high quality images to be obtained up to 200 m. The Mesozoic bedrock was a distinct and very strong reflector in the resulting reflection seismic imaging. There was also a clearly visible depression of the horizon corresponding to erosion or a structure (syncline). Deeper, it was possible to track two or even four detachments of faulting origin.

**Keywords:** reflection seismic imaging; multi-channel analysis of surface waves; first-arrival travel-time tomography; Mesozoic bedrock; multi-method analysis

## 1. Introduction

Recognition of the Mesozoic basement in the Holy Cross Mountains and the tectonic structure lying below was the main aim of this paper. During geological research, the use of geophysical methods is invaluable, especially the use of high-resolution seismic methods [1–6]. These methods image geological structures at different depths with different degrees of precision [7,8]. Seismic methods have been successfully applied in the exploration and exploitation of hydrocarbon deposits and actually in many other fields of science, for example, archaeology, construction, environmental protection, tectonics, and geomorphology [9–16]. Another application worth mentioning is sediment basin analysis [17–23].

Geophysical methods provide the only opportunity to obtain information about subsurface structures in areas with poor geological knowledge, without deep boreholes and outcrops, in both 2D or 3D. The technical solutions (e.g., source type) used during fieldwork and acquisition geometry determine

the quality of the seismic data and the depth of investigation [24]. Therefore, the combination of different complementary seismic methods may result in more consistent, accurate, and detailed interpretations.

To fulfil such a demand, multi-channel analysis of surface waves [24,25] and first-arrival travel-time tomography [26] were used along with seismic imaging. This research used the uncertainty-driven approach proposed by Marciniak et al. [27], which utilizes all possible data from various geophysical methods to accomplish a more precise and clearer result. This kind of approach requires estimation of uncertainty at each step of processing to obtain a reliable interpretation of the geophysical image as the geological result. By applying geophysical methods in sequence, where less certain methods are used at the beginning and the most precise are used at the end, and transferring information between methods, the authors were able to increase the quality of the seismic stack. Considering a few geophysical methods by using the multi-method approach gave a more detailed and accurate image of the basin.

## 2. Geological Setting

The research area is located approximately 5 km from the village of Chęciny in the Holy Cross Mountains, between Grzywy Korzeczkowskie and the rivers Hutka, Biała Nida, and Czarna Nida (Figure 1). The Mesozoic bedrock contains Upper Jurassic deposits, mainly Kimmeridgian [28–30]. The Upper Kimmeridgian deposits include shell limestone, oolitic limestone, marlstone limestone, and marl, as well as marly clays. The Lower Kimmeridgian deposits consist of oolitic limestones, chalky limestones, pelite, and striped limestone with flints [30]. The nearby hills are made of rocks belonging to the Lower Kimmeridgian, namely Grzywy Korzeczkowskie (over 330 m a.s.l.) and Bzowica Mountain (240 m a.s.l.) on the other side of the village of Mosty. In the vicinity of the studied area, there are also Oxfordian rocks. They include speckled limestones, chalky and marginal limestones, and limestone with flints. In some places they reach the surface, where they form the northern and eastern parts of Grzywy Korzeczkowskie [30]. The faulted and folded rocks described are a direct bedrock for Quaternary sediments.

The axial part of the Bolmin syncline is located in the Mosty region [28,29]. Its axis has a WNW direction in this area. In the SW area of the Holy Cross Mountains Margin, a number of fault zones have been found, which are partially visible in the relief of this terrain (Figure 1). Most river valleys, including Hutka, Biała Nida, and Czarna Nida, are characterized by developed fault lines [28,30,31]. The largest fault zone stretches along a length of 15 km to the NW through the Mosty region to Czarna Nida to the south of Chęciny. It cuts the rocks of the Bolmin syncline and has the character of a dextral strike–slip fault [32,33]. The existence of smaller faults can be suspected on the basis of interpretation of the relief of Grzywy Korzeczkowskie (Figure 1). Gorges that cut the hill have been formed as a result of the tectonic activity. In this part of the Holy Cross Mountains, the relation of the relief with the tectonics and lithology of Quaternary bedrock is exceptionally readable [32].

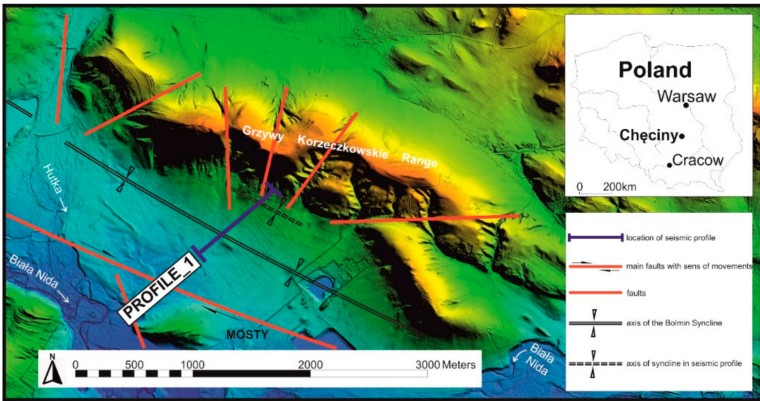

**Figure 1.** The studied area, Chęciny in the Holy Cross Mountains. A fluvial area (with the sand mine) is located between culminations of Mesozoic rocks (Lindner et al., 2001; Cabalski et al., 2018).

In the Mosty region, the flat surface to the SW of Grzywy Korzeczkowskie is built of Quaternary sediments. The thickness of the Quaternary overburden is estimated to be up to 90 m in this area, but is usually at the level of 30–60 m. The profile of the Quaternary sediments begins with a layer of debris and residual clays classified as the South Polish Complex [32,34]. Sand and gravel sediments of the Middle Polish Complex fill the fossil valley [35], the bottom of which was recognized in boreholes at a depth of at least 140 m a.s.l. to the east of the village of Mosty [32]. Above it, silts with an admixture of sands appear. Their genesis is associated with obstruction of the surface runoff by the forehead of the Odranian-age glacier, which is located approximately 15 km to the SW, near Małogoszcz and Łopuszna [33,35]. In the immediate vicinity of the seismic profile, these sediments occur at depths of about 30 m [36,37]. The thickness of the series can reach several metres [32]; however, it is usually smaller as a result of subsequent erosion of proglacial and extra-glacial waters. The accumulation of sand–gravel material with a thickness of 10–25 m is also related to the activities of these waters [36,37]. These sediments, which directly form a large area between the foothills of Grzywy Korzeczkowskie and the village of Mosty, represent a higher level (V) of the Biała Nida valley [38].

The lithology and texture of subsurface sediments can be traced in the walls of the nearby sandstone in Mosty. Within this layer, there are packages and interbedding of fine debris of diluvium origin on several levels. These paleoflow sediments from the slopes of Grzywy Korzeczkowskie were accompanied by the accumulation of extra-glacial rivers in periglacial conditions. The debris layers can be several metres thick, and their occurrence is irregular both in the profile of sediments and on the surface [37]. In the SW direction, the surface is made of the sands and gravels of the Biała Nida River terrace, created during the main Vistulian glaciation (Lindner et al., unpublished data). Despite being a continuous object of geological interest, the general knowledge about the Holy Cross Mountain is still incomplete. Many geological hypotheses still need to be verified, especially using seismic methods.

## 3. Fieldwork

Two seismic profiles were obtained during the survey. The analyzed one (Profile_1, Figure 1) was 840 m long with a receiver spacing of 5 m and a shot spacing of 2.5 m. Sixty standalone GPS-based DATA-CUBE recorders with 4.5 Hz geophones were used as seismic receivers. The source, an in-house modified weight drop (PEG-40), produced enough seismic energy to image the studied structure. Offset shooting with 10 shot points on each side was carried out along five deployments of Profile_1. This gave a good, stable seismic fold of 90 on average, without significant oscillation. To meet the need for precise topography for all methods, geodetic measurements were conducted using an GPS RTK LEICA device. Accurate shot timing was ensured using GPS-based devices, which were specially constructed and had been previously tested for geoengineering problems [8]. The gathered data (Figure 2), stacked using a diversity stack, were used for a few seismic methods: first-arrival travel-time tomography (FATT), multi-channel analysis of surface waves (MASW), and reflection seismic imaging. The presented data gathering scheme allowed for fast and cost-efficient acquisition.

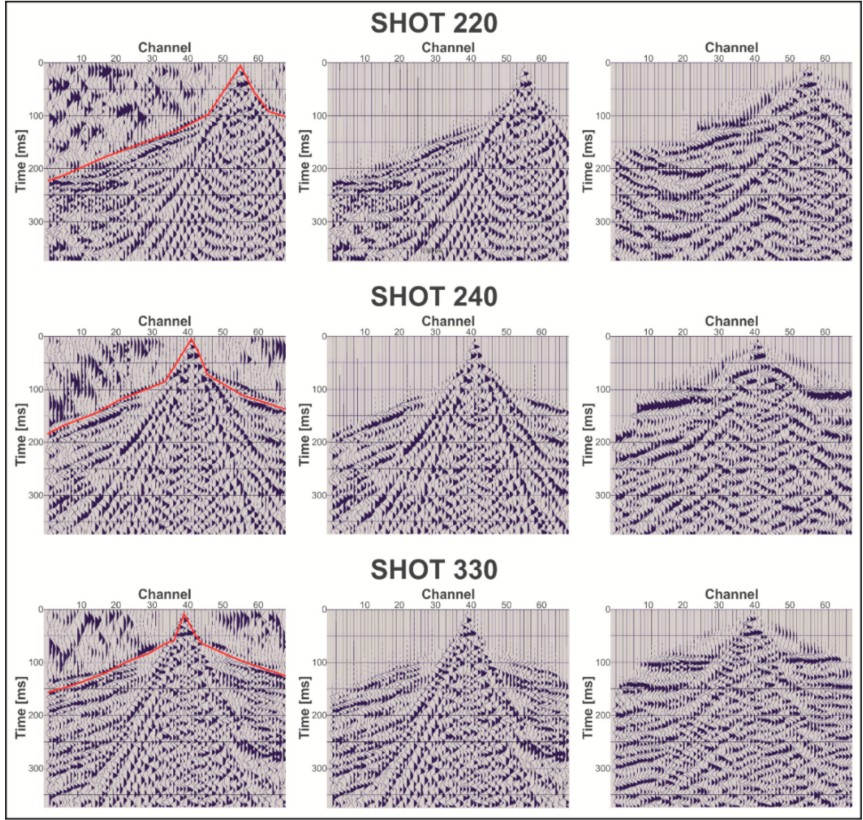

**Figure 2.** Shot gathers for shots 220, 240, and 330 with applied automated gain control (AGC) (100 ms). Left panels: shot gathers after diversity stacking. A carefully designated front mute (red line in left panels) removing linear refraction arrivals is presented in the middle panels. Right panels show the effect of applying normal move-out (NMO). Clear shallow reflections were visible at approximately 120 ms for shot 220, 110 and 140 ms for shot 240, and 100 and 150 ms for shot 330.

## 4. Methods and Data Processing

In the multi-method analysis, the order in which methods were used was not accidental, but dependent on their uncertainty and depth of investigation [27]. In the first step, the shallowest and most uncertain by resolution method, MASW, was used to identify the shallowest structures. The obtained results were used to create a preliminary model of the velocity field for FATT and for static correction. FATT was then applied to estimate the velocity field for refracted waves. In the last step of the multi-method approach, the most precise and demanding method was used: reflection seismic imaging, which utilized all of the information from MASW and FATT to give better constraints and more robust results.

Each seismic method demands a different processing scheme. However, the methods have a few common steps at the beginning. Geometry was set up and quality control was performed. These processes take a long time but are crucial to obtaining a high-quality result in all methods, especially due to the use of standalone stations and possible acquisition errors. To remove noisy channels, manual trace editing was carried out. A vertical stack of four repeated shots was performed to improve the signal-to-noise ratio.

### 4.1. Multi-Channel Analysis of Surface Waves

Both tomographic and reflection methods are poorly sensitive to potential low velocity zones, and also cannot image upper 10 m of the shallowest subsurface; therefore, the MASW method was applied. This analysis, widely used in geoengineering projects, can provide information about the

shear velocity (Vs) field [24]. Furthermore, it can be used to estimate the geotechnical parameters of the soil.

In the presented study case, a scheme where multiple single 1D profiles were interpolated to obtain a pseudo-2D model was used. The preprocessing of the data, performed using Globe Claritas software, aimed to enhance the surface waves and suppress the noise. Particular attention was paid to the removal of refraction and reflection waves from the records; these can potentially strongly affect the result. That goal was achieved by applying a combination of a 1D band pass with characteristics of 2–3–60–70 Hz and 2D FK filtration, followed by 500 ms automated gain control (AGC). The later steps of processing were standard for the classical MASW scheme [39]. The dispersion curves were picked manually and properly prepared for the inversion procedure. To obtain the best possible input data for inversion, every dispersion curve was picked 10 times, and average of them with the estimated misfit as standard deviation was used in the modeling (Figure 3). Moreover, in some noisy single shot records, the picked curve was smoothed to eliminate the impact of hand precision as much as possible. To obtain the 2D model of Vs, nine single 1D dispersion curves at points spaced at regular intervals along the seismic line were analyzed. The inversion procedure was based on the Monte Carlo and the Nearest Neighbourhood algorithms adapted to the Dinver mode of the Geopsy program [40]. The starting model created took into account information from LVL (low velocity layer) analysis and local geology. During the inversion, 50 Monte Carlo models and an additional 2500 models from the Nearest Neighbourhood algorithm were used to find the one with the best fit to the data.

As an overall uncertainty criterion, the chi-squared parameter was used. Results below 1.3 were rejected. Additionally, the theoretical dispersion curve was compared manually with the original one for each of the nine locations as a final verification. Interpolation between profiles was achieved by the kriging method in the Surfer program. Finally, a smooth 2D model showing the general trend of changes in lithology for the first 50 m was obtained (Figure 4).

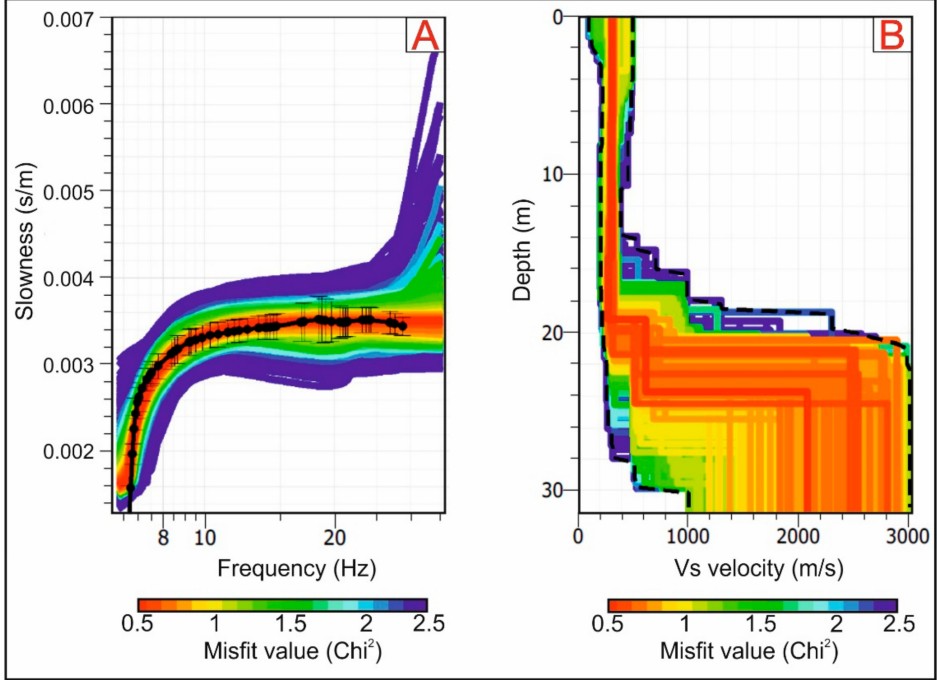

**Figure 3.** (**A**) The uncertainty of the dispersion curve modeling. The picked dispersion curve (black line) with the uncertainty of picking was well fitted to the synthetic models (colored areas). The chi-squared parameter for the best theoretical curves was less than one; (**B**) 1D result of the multi-channel analysis of surface waves (MASW) analysis for shot 164. The 1D velocity model presented a uniform structure up to 18 metres. Below that depth, an increase of shear velocity (Vs) was visible. Interpretation of the deeper parts was impossible, due to the penetration range of the method.

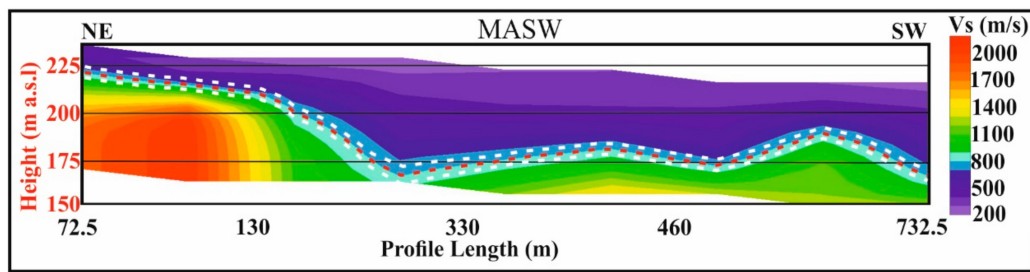

**Figure 4.** The result of the multi-channel analysis of surface waves (MASW). The red dashed line presents the geophysical boundary. The white dotted lines show the uncertainty of the result. The overall shape of the structure is similar to the result obtained in reflection imaging.

### 4.2. First-Arrival Travel-Time Tomography and Uncertainty Analysis

The velocity model spanned a rectangular grid with $85 \times 31$ cells and dimensions of $10 \times 5$ m. An inversion scheme consisting of 10 steps with six iterations per step was applied using JIVE3D software [41]. JIVE3D uses the shooting method [42] to trace ray paths through a modeled velocity field. The linearization of the relationship between travel-times and model parameters was carried out using ray perturbation theory. The inversion method was based on a set of linearized refinements to the initial velocity field to achieve a better match between the starting model and the raw data.

Clearly visible at all offsets, P-wave first-arrival travel-times were picked manually along the profile. Similarly to Korenaga et al. [43], a large number of starting models were tested to find the optimal one—that is, the one which gave the best fit to the travel time at a given parameterization. Instead of Monte Carlo sampling, a simple grid search approach was adopted. Our starting models were generated by a set of 1D linear vertical velocity gradients with defined topography. Each gradient was parameterized with two values: the velocity at the surface (V1, above topography) and the velocity at the bottom of the model (V2). Interpolation of the velocity value for each model's cell was performed using standard b-spline methods. Table 1 summarizes the gradient parameters with the corresponding steps that were tested.

**Table 1.** Range of values of the upper (V1) and lower (V2) velocities in the experimental setting.

| Velocity | Elevation (m) | Minimum (m/s) | Maximum (m/s) | Step (m/s) | Final Model (m/s) |
|----------|---------------|---------------|---------------|------------|-------------------|
| V1 | 300 | 200 | 300 | 10 | 260 |
| V2 | 150 | 1200 | 1800 | 60 | 1500 |

In total, 121 starting models were found to have adequate statistics for evaluating the uncertainty using hit-count normalization [44]. In this approach, the standard deviation normalized by the sum of rays in each cell was treated as the uncertainty estimator, called a weighted error (WE). Inversion was carried out for all 121 starting models of the velocity field. To find the optimal velocity field, the travel-time fitting was estimated as a ratio of chi-squared to the hit-rate—that is, the percentage of the picked travel-times for which synthetics were successfully obtained during two-point ray-tracing. The best travel-time fit was obtained for the model where V1 = 260 m/s and V2 = 1500 m/s. The inversion of this model was considered to be the final result of tomography. In the next step of the multi-method analysis, it was used as the subsurface velocity field and for correct front mute in reflection seismic imaging. Figure 5A presents the velocity distribution from inversion using JIVE3D code with the mask. The velocity field was rather smooth. The unconfirmed area (without sufficient ray coverage) has been greyed out.

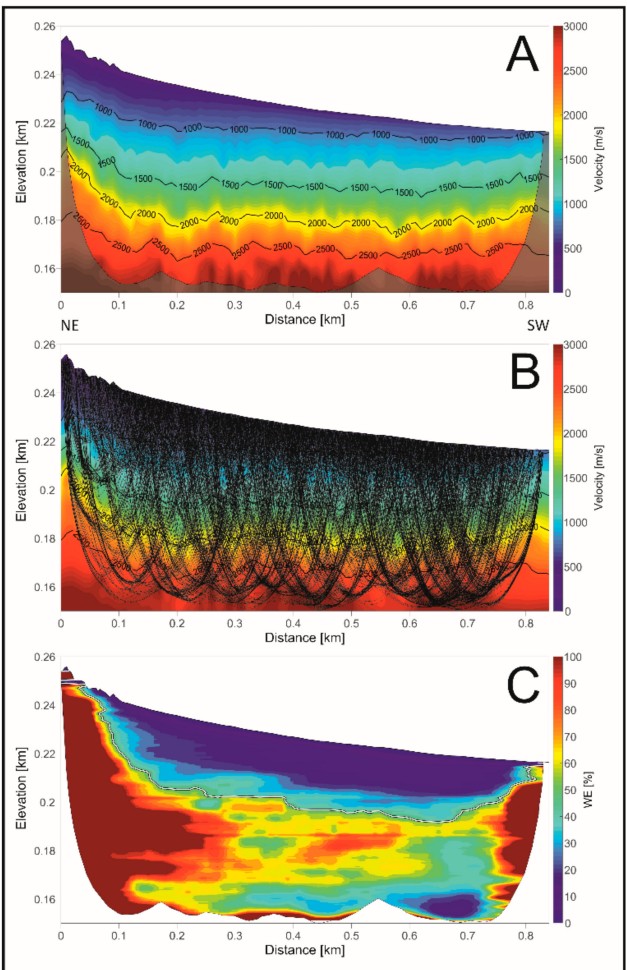

**Figure 5.** The best result of the first-arrival travel-time tomography (FATT): (**A**) The velocity field. The unconfirmed area without ray coverage was greyed out. The color contour corresponds to the resolved part of the result with high-density ray coverage; (**B**) ray paths; (**C**) the map of uncertainty normalized by the ray coverage – WE (weighted error). The black and white line corresponds to the edge of the area well-resolved by rays (in this particular case, a value of 50%).

The estimated uncertainty is presented in the form of uncertainty maps (Figure 5C). Each cell of these maps was calculated separately as the standard deviation of the velocity from the respective cells, extracted from the models. Masks in Figure 5A were created considering only those parts of the given panels covered by rays (Figure 5B). Hit-count normalization in Figure 5C did not require masks because it occurred automatically, due to the nature of this procedure.

The FATT result showed an almost-flat smooth structure slightly inclining to the SW. There were also visible small-scale undulations in the shallow part of the section.

### 4.3. Reflection Seismic Imaging

With a limited offset of 100 m, refraction statics were estimated by applying standard techniques from Globe Claritas. The seismic imaging processing also included a carefully designated front mute (red line) to extract linear refraction arrivals without distorting wide-angle shallow reflections (Figure 2, middle panels). Its meaning has been demonstrated in many articles [45,46]. The outcome of the NMO (normal move-out) correction is shown in Figure 2 (right panels) as flattened shallow reflections. Straightening of reflections confirmed the correctness of the velocity field used. The appearance of strong surface waves is problematic for seismic imaging. They interfere with wide-angle reflections, which are treated as a useful signal. A simple high-pass filter with frequencies of >30 Hz proved to

be sufficient to suppress almost all surface wave amplitudes. In the next step, a surface-consistent deconvolution was performed in order to improve the resolution of the seismic image by amplifying high frequencies. The data were filtered in the FK domain using a band pass filter with characteristics of 2–3–100–120 Hz before stacking the shot gathers in the seismic section to cut frequencies above 100 Hz. Final time to depth conversion was performed using the velocity model from the previous FATT and further extrapolated downward with the standard common velocity stack (CVS) method. The final result in the depth domain is shown in Figure 6B.

Due to the lack of boreholes, verification of the interpreted structures was impossible. To evaluate the uncertainty of the main reflector determination, the velocity field used for the time to depth migration was changed by applying extreme values of velocity. These velocities were estimated for ±30% of the optimal one according to knowledge from MASW, FATT results, and previous works [7]. Figure 6C shows the estimated uncertainty in the form of white bars with the geological interpretation.

The reflection imaging provided the possibility of recognition of the Mesozoic basement visible as a strong reflector. Application of the proposed approach gives information about the uncertainty of the result.

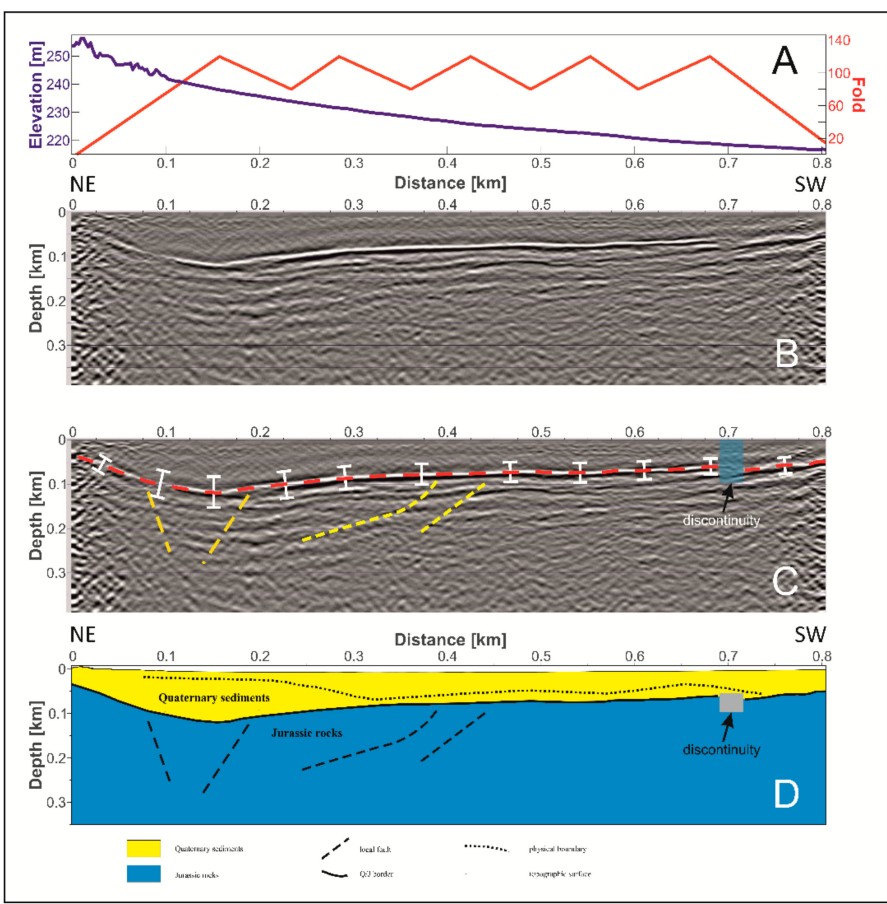

**Figure 6.** The result of the reflection seismic imaging: (**A**) Elevation and fold. Thanks to acquisition geometry in the form of six deployments, a good seismic fold along the whole profile was achieved; (**B**) seismic stack in the depth domain; (**C**) interpretation of the seismic image. Considered as the Mesozoic bedrock, the main reflecting horizon (red dashed line) is clearly visible in the seismic section. Its depression in the NE is probably related to the syncline structure and/or erosion. In the middle of the seismic stack (yellow dashed line), there are four detachments, probably connected to a local fault. White bars show the estimated uncertainty of determination of the interpreted horizon. The blue area corresponds to the discontinuity of the Mesozoic bedrock; (**D**) geological interpretation of all results.

## 5. Discussion

The presented model of Quaternary sediments indicates that regardless of origin and age, their lithological variation is small, with the exception of the thick packages of limestone debris. This issue certainly hinders their spatial recognition and interpretation in seismic results [36,37]. The pronounced contrast between the physical properties of Quaternary sediments and those of the Jurassic rocks was the main basis for undertaking the research.

The result from MASW showed simple flat-lying structures according to shear velocity to a depth of approximately 40 m, but smoothly inclining to the SW (Figure 3). Therefore, static correction was not required for this dataset. The FATT result showed an almost-flat smooth structure with small-scale undulations slightly inclining to the SW. The places where velocity decreased are related to thicker deposits of loose material (Figure 5A, at approximately 120 m). The uncertainty map presented in Figure 5C was computed as shown in Owoc et al. [44]. The well-resolved part of the velocity field, which is crucial for the most critical processing step in the seismic imaging, the manual front mute, is visible as the upper bluish zone with the lowest values of WE. During stacking of the seismic data and time to depth conversion of the final stack, this part of the velocity field was also used.

As shown in Figure 6B, the obtained image was not distorted by any amplitude variation and was therefore easy to interpret (Figure 6C,D). This feature was due to an almost uniform seismic fold (Figure 6A). At a depth of approximately 50 to 120 m, the main reflecting horizon was clearly visible due to the high contrast of impedance between the soft Quaternary sediments and the hard Jurassic rocks (Figure 6B–D). This reflector was interpreted as the Mesozoic bedrock. It definitely inclined to the NE direction, where there was an important feature: a depression at approximately 120 m of the profile with its bottom around 120 m below the subsurface. From this location on the profile in both directions, this horizon rises to 40–50 m depth. In the SW part of the seismic result, the Mesozoic bedrock became almost flat. In the vicinity of 700 m of the profile, the signal suddenly weakened, which was visible as a rapid discontinuity of the reflector in the result of the reflection seismic imaging. In general, the shape of this line referred to the gently curved syncline with the axis in the northern part of the presented seismic profile (Figure 6B,C,D). This would mean that the axis of syncline should be moved several hundred metres to the north compared to the previously developed interpretation [32]. However, the gentle nature of the Quaternary basement line suggests a glacial erosion. The glacier adapted to the existing shape of the syncline bedrock, deepening the axis zone and leveling the adjacent areas. The first glacier in this area during the Sanian 1 (MIS 16) glaciation [28,41] entered with a lobe from the SE. A slight residue of glacial tills proved the small thickness of ice in this region [28]. Therefore the results of the glacial erosion may not be significant. The river erosive genesis seems to have had a narrow and deep refraction of the basement line in the southern part of the seismic image. The existence of fossil river valleys from the Mazovian interglacial was documented by Lindner et al. [32] in the SE region of Mosty. Perhaps it is a continuation of this valley form, although the bottom of visible discontinuity (valley) on the discussed profile reaches about 100 m from the surface.

The seismic signal below the line interpreted as the Mesozoic bedrock allowed two types of reflections to be recognized. The first ones were gentle lines or sets of lines approximately parallel to each other and with the same shape as the basement. The second type consisted of steep lines or interruption zones of the seismic signal (Figure 6B). It is very likely that the bending of the lines reflected the variation of sediment layers within the axial part of the Bolmin syncline. Its inflection point corresponded to the intersection of the cross-section with its axis. The syncline axis was located near the foothills of the Grzywy Korzeczkowskie within marly Kimmeridgian limestones [30]. The visibility of the syncline layering disappeared below the depth of 350 m. This was probably related to the occurrence of signal-suppressing rocks: clays, claystone, siltstones, and the Lower Jurassic and Upper Triassic marly limestones. The steep lines in the seismic image were identified as discontinuities of rock layers, locally with vertical displacements—faults. Four detachments were found under the main horizon in the middle of the seismic image: two in the middle part of the profile and the other two in its northern end, accompanying the inflection zone of the syncline. They correlated well with the

faults that cut through the Grzywy Korzeczkowskie Range (Figure 1). The extensions of the fault lines coincided with the seismic record discontinuities. The inclination angles of the upper sections on the discontinuity line (apparent dip) were from approximately 40° to 55°. Considering the intersection angle of the fault line on the plan with the section line gave the approximate value of the fault slope in the range from 51° to 65°.

The diversification of the Quaternary sediments in the seismic image was definitely less spectacular. Signal gain zone, parallel to the surface, was recognized at a depth of 5 to 20 m, especially in the NW part of the profile. The second lay at a depth of 30–90 m; however, it sometimes disappeared. These signal amplifications could have been the result of reflection from layers with a large admixture of limestone debris or increased content of gravel. Zones without clear reflections can be interpreted as attenuation of the seismic energy by silty and fine-grained layers.

The obtained seismic image generally confirmed the current interpretation of the geological structure of the studied area, but structures are at different locations (Figure 6B–D). However, it provides new details in several places. The synclinal arrangement of the layers was confirmed; however, the axis of the syncline should be located in the area of the southern foothills of the Grzywy Korzeczkowskie. In the case of strongly faulted terrain, local changes in the course of the syncline axis do not arouse controversy even in short sections. The shape of the sub-Quaternary bedrock was in line with the geological structure of Mesozoic rocks. Namely, the lowest surface is located on the axial part of the syncline. Strong tectonic involvement of this area, visible in the morphostructural analysis, has also been confirmed. The faults reach the ceiling surface of the Mesozoic rocks and are not visible in the quaternary overburden. However, their relationship with a deep and narrow erosion form in the Mesozoic ceiling surface at the southern end of the cross-section can be found indirectly. The Quaternary bedrock, in addition to the exception described above, was imaged on the seismic profile by a strong but gentle line. This is the result of not very intense glacial erosion rather than time-differentiated river erosion. Only the discontinuity zone in the SE part of the profile may have been associated with deep river erosion, as previously suggested by Lindner et al. [32] and Lindner and Mastella [33]. The thickness of the loose rock cover reaching a maximum of 120 m was confirmed.

The usefulness of shallow high-resolution seismic profiling in geological studies can be considered significant. However, separation of Quaternary deposits was impossible. This can be explained by the relatively weak lithological variation within the sedimentary cover and the small thickness and irregularity of the occurrence of rubble packages, giving a stronger seismic signal.

## 6. Conclusions

Relatively high-quality images of the subsurface with low cost of acquisition make seismic methods cost-efficient and desirable to researchers. The multi-method approach, which allows the utilization of all recorded wave fields (surfaces waves for MASW, wide-angle refractions for FATT, vertical reflections for seismic imaging), provided detailed images with high resolution down to 200 m. During acquisition, the timing device and the in-house modification of the seismic source were working properly. Combined with good and dense coverage of the studied area by seismic shot and receiver points, the multi-method approach gave well-resolved and detailed images of geological structures. The seismic energy obtained from the in-house modified weight drop during acquisition was strong enough to image the structure down to 200 m. The extra time spent on MASW processing and seismic tomography in comparison to a standard approach using only the single method of seismic imaging was worth the effort.

The strong main reflector at depths from approximately 50 to 120 m was connected with "hard" (Mesozoic) bedrock. Its shape refers to the syncline structure of the basement and erosional processes. The axis of the syncline is located near the foothills of the Grzywy Korzeczkowskie. The flattened shape of the bedrock in the studied part of the Holy Cross Mountains may be indicative of the limited erosional activity of the glaciers. In the NE, a narrow and deep depression form was found in the Mesozoic bedrock. This structure is probably related to river erosion during one of the

interglacials. In the Mesozoic rocks, four detachments with a steep inclination, probably related to local faulting, were identified. Their configuration is associated with the main surface morpholineaments. The physical limitations of reflection seismic imaging did not allow the precise imaging of the first thirty metres.

**Author Contributions:** Conceptualization, all authors; methodology, B.O., A.M. and M.M.; validation, all authors; formal analysis, B.O., A.M. and M.M.; writing—original draft preparation, B.O., A.M., and J.D.; writing—review and editing, all authors; visualization, all authors.

**Funding:** This research was funded by NCN Grant UMO-2015/19/B/ST10/01833, entitled 'Three dimensional model of the lithosphere in Poland with verification of seismic parameters of the wave field'. Part of this work was supported within statutory activities No. 3841/E-41/S/2018 of the Ministry of Science and Higher Education of Poland.

**Acknowledgments:** We are grateful to M. Ludwiniak, for a fruitful discussion on the tectonics of the study region.

**Conflicts of Interest:** The authors declare no conflict of interest

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
