# Peer review of "Seismic Imaging of the Mesozoic Bedrock Relief and Geological Structure under Quaternary Sediment Cover: The Bolmin Syncline (SW Holy Cross Mountains, Poland)"

_geosciences, doi:10.3390/geosciences9100447_

Round 1

Reviewer 1 Report

The paper entitled “The Mesozoic bedrock relief and geological structure under Quaternary Sediment cover in a seismic investigation: the case of the Bolmin syncline at Mosty site (SW Holy Cross Mountains, Poland)” by Owoc et al presents shallow seismic structure in SW holy Cross Mountains Poland, with applying multi methods, such as seismic travel-time tomography, seismic reflection survey, MASW and GPR. The authors have successfully obtained fine structure in the shallow portion in the target area. Overall, this paper is well written as a technical report. However, the paper, in its current form, includes some of lengthy parts in the text, especially in the section 4, “Methods and Data Processing”. Additionally, all of paragraph has described wordy, and fail to mention clearly their points in each paragraph. I strongly recommend that the editors accept a revised version of this paper after re-editing the current version with small paragraphs with showing a point in each paragraph clearly.

Author Response

Reviewer suggestions have been applied to manuscript.

Reviewer 2 Report

Review “The Mesozoic bedrock relief and geological structure under Quarternary sediment cover in a seismic investigation: the case of the Bolmin syncline at Mosty site (SW Holy Cross Montains, Poland)”

This is a quite well-written contribution describing a shallow multi-seismic method approach along a short profile in the Holy Cross Mountains, Poland.

I think the paper is usually clear, structured and complete. However, I do fail to see the significance of this study, and this might be that the paper is pitched as a methodological paper (I think), instead of focusing on the geological problem, which may be very interesting for geologists. The combination of methods is nothing new and details of how this supposedly new approach works and what makes it new are missing. At the moment, I don’t see this paper neither as a full methodological paper (which seems to have been submitted elsewhere, Ref 27) nor as a good presentation of the geological implications – it is somewhere in between. Perhaps the authors could decide which way to go and focus the paper more on either of these directions. I find the geological implications interesting, but they would need to be described in much more detail, as well as the implication for the geological problem and their significance. How do the results change interpretations of Quarternary(?) geology and models in the region?

Perhaps the authors are able to give this contribution a little more straight-forward and conclusive overall story/picture.

Although I said that this paper is sound, generally complete and well-structured, I feel that changes need to be made. I will label them as "minor revisions", but I think it is a little more than that. I know my requests seem a little unclear and vague, but I would ask the authors to try to focus on these comments, which will potentially make this paper a  great contribution.

General remarks:

Please make a summary figure of all the basement interpretations and structures from surface waves, tomography and reflection seismic beneath the topography. Just as line-drawings of the horizons, faults, etc.

Is GPR part of you methods or not? I feel you said so, but there is essentially nothing documenting or describing GPR results.

Detailed comments:

L2-5: Title, quite long. It’s not really extremely important, but a shorter title would perhaps be nice, an idea to shorten: “Seismic imaging of the Mesozoic bedrock relief and geological structure under Quarternary sediment cover: the Bolmin syncline (SW Holy Cross Montains, Poland)”

L13: cross “current” – I think there was always popularity, maybe even more back in time?

L13-15: I know what you mean here, but I think it is not entirely correct. The quality of geophysical results – what do you mean? The raw data? The interpretation? I am not sure of you talk about the quality increases with stacking of data? Or do you mean the interpretations get better by combining data? How would the use of several data improve the resolution (i.e. how would the shar wave analysis improve the resolution for the tomography?). Do you mean resolution (as in seismic)? Can you not just simply say that “analysing several complementary geophysical/seismic datasets allow an improved interpretation”, or similar?

L21: “significant improvement in the result” – again, what do you mean by result, the interpretation of all analyses or results in general (the individual images)? Please be more specific?

L21-22: I would argue that the individual images have good quality/resolution up to 200 m depth not because the joint interpretation, but because of the quality of the data.

L22-23: “the result of the seismic imaging” – sounds awkward. Also do you mean all images or certain ones? Why not then just say “in the seismic images”?

L32: “seismic methods/techniques” instead of “the seismic ones”. “which give the most precise results”. Well, I don’t think you can say that so easily. Also, compared to what? Depends on the methods, gpr could be more “precise”, CT/MRT are more “precise”. Just say “especially the use of high-resolution seismic methods”?

L33-34: “Thanks to constant improvements”… I think this is unnecessary and also – how much have they improved in the last 30 years? Seismic methods have been used for all named applications for decades. I would start the sentence: “seismic methods are successfully applied in …”

L36-38: Fine, but this is also nothing new… decades of research of sedimentary basins and structures with “geological research” applications.

L42: I would rephrase: “Combination of different complementary seismic methods may result in (or may obtain) more consistent, accurate and detailed interpretations.”

L43-45: “to fulfil such a demand…” Mention that you dd this “in THIS study”, or use “we used it”, still sounds like part of a general statement.

L46-50: The reference “Marciniak et al” is submitted, not published. That’s OK, but then you ened to go into a little more depth describing the principles of this method. How does it work, what does it involve? A step-bystep analysis of each dataset, in which order?, or does it work iterative (do later results feed back to earlier models?) a joint analysis? Please add a few more sentences. At the moment it doesn’t really say anything but that it is a multi-method approach with careful analysis and this results in a clearer result.

L47: What’s the sub-surface zone” or better: What is not the sub-surface zone? All results should surely be in the subsurface, so why “especially” implying that there is another zone.

L47-48: I would say every approach requires a careful processing… which “steps”? Please add details how the method works (see also above comments).

L54: “Quaternary bedrock”??? In what way is this Quaternary?

L64: What “tectonic effect”? You mean the bedrock morphology has an effect on the deposition of Quaternary sediments?

L67: How can river valleys be “characterised by developed fault lines”? How characterised? Did they form along faults?

L67: Noidea where Malogoszczcza and Czarna Nida, Tokarnia are.

L71: I would say the smaller faults can be inferred from relief interpretations, but this is not proof. Can they actually be seen in the geological record? That would be proof.

L73-74: Rephrase.

L74-76: “convergence” of relief, tectonics and lithology sounds odd. Maybe “relation” is better?

L85: “the bottom of which” instead of “which bottom”.

L86: “above,…”, above what? Above the bottom? Maybe “the lowest level of the basin infill”? What is “slits” – “silts” or siltstones?

L89: How is the depth of these sediments know? Geological mapping/outcrops? Boreholes?

L104: “is circa 840 m long…”?

L109: “geodetic measurments”

L110: “using an RTK LEICA device” – what kind of device is that?

L113: explain abbreviation MASW (like for FATT vbefore).

L124: Why the most uncertain? In which way? Do you mean the methods with least depth penetration?

L127: Perhaps to “estimate” and not “recognise”?

L131-132: There’s something from with the sentence “Firstly….” Please check and rephrase.

L133-134: what does “geometry was added”? Also, “quality control was performed”, not added.

L139: what are zones with “lower velocity”? Sure, they can image low velocity zones/structures/anomalies, at certain depths and amplitudes? What exactly do you mean?

L140: “upper 10 m” instead of “first 10 m”.

L15: “average” not “middle”. “How “with the estimated misfit”? As a prior uncertainty? As a weight in the averaging? The actual misfit or also the “range” (i.e. standard deviation) of different dispersion curves?

L155-157: References for inversion methods missing.

L157: So you create a starting model, OK. How does the model utilise LWL analysis (and what exactly is this) and how the local geology? Did you perhaps misuse “unitlise”?

LL161: A comparison is done (very awkward) -> they are compared.

L173: Please spell out the abbreviations once more for each figure (also in L202).

L177: How were the GPR data analysed for the surface wave modelling? Doesn’t seem like a joint inversion. So please explain. Also, if GPR was used as a prior for surface waves, it should be described first.

L181-181: What does “no impact on seismic reflection” mean?

L176-182: You need to add more description of the method, analysis, show figures, etc. There is no documentation of this or the results. If you in the end don’t use GPR, then don’t describe it.

L184-185: A grid with NxM cells and dimensions of AxB m.

L186”this parameterisation was defined after initial tests and is assumed to give optimal results”?

L190: What exactly are “linearised refinements”?

L191-192: Would be good to see one example of picking and the subsequent data fit.

L207: “From” and “to” sound quite strange. Maybe minimum and maximum tested values.

L217: “seismic imaging” is vague and includes everything you did before.

L216-218: I don’t understand this sentence.

L227-228: Difficult to understand what you men in this sentence.

L231-233: Again, something wrong with sentence. “The outcome of the NMO procedure is shown…, conforms”. You cannot continue with “conforms” in this sentence.

L231: Is this NMO procedure a NMO correction?

L236: You could write “surface wave-amplitudes”

L237: in case of “surface consistent deconvolution”, why use “the” and not “a”?

L 238 “and using a band pass…”, no “and”. Previously you filtered >30 Hz, why both?

L240-241: Again, quite inconvenient: Why not just say: “depth conversion was performed using the velocity model from the previous FATT”?

L244-248: OK, but why didn’t you use the observed uncertainties (or does 30% represent this)?

L259-260: “The presented description” – why description? Would “models” not be a better description of what you did in this work? Also, you can actually only say that the seismic velocity variations are small. From that you may go a step further and interpret that the lithological variations are also small. Another thing, I think you mix up your results and previous knowledge in this statement. OK, the seismic velocities suggest a pretty homogeneous sedimentary package, but where does your conclusion about the limestone debris come from? You haven’t mentioned this in the description of your seismic results.

L264-265: “…show simple, flat-lying shear-eave velocity structure to a depth of approx. 40 m.”?

L265-266: There is no radargram shown. Add! How can small-scale variations in GPR have an effect on the seismic processing? Please explain. Do you mean that could correlate with velocity perturbations? I don’t understand.

L264-273: General comment: Yes, quite flat-lying structure, but don’t forget to describe that it is smoothly inclining to the SW!!!

L267-268: Where are the “zones of lower velocities”? Cannot follow this description. Would a description of “generally flat-lying horizons with small-scale (meter-to 10m-scale) undulations” not cover this better? So this actually means that the basement is not really smooth!?! Also this variation is not really seem in the reflection image – is it that you want to explain in the following – resolution in the FATT is poor?

L275-277: Sorry, I have difficulties to understand what the “fold” in Fig.6A is supposed to tell and where it comes from. Also, where is the supposed fold in the seismic image?

L278 …: What does “falls into the NE direction” mean here? L280: “depth” not “deep”?

L279: This depression coincides with a depression in the tomography, but not really surface waves – I think. Please add a figure comparing all results.

L282: So what is this discontinuity?

S285: What’s the “previously developed image”? Where can I see it? Do you mean “compared to previous interpretations”. Can you show this “image” in comparison to your results?

L286-289: I fail to understand the meaning of this sentences in context of the previous sentences. There may also be grammatical mistakes which makes it difficult to understand what you mean. Please rephrase.

L291: Which “main glacier mass”? How do you know?

L286-297: This whole section is quite cryptic in my view. Please try to make this more clear.

L300: “co-shaped”?

L306: What as “resistivity” (less-resistant rocks) to do with the seismic signal?

L312-313: rephrase: “discontinuities in the seismic record”.

L316: What exactly do you mean by “subsurface zone” ( do you mean the quaternary sediments?) and why is it “less spectacular” – that sentence doesn’t add any information. Please be more specific what you mean.

L316-323: Seems to me that there is also a lot of noise – hard to see (for me) what is structure and what not.

L323-324: Didn’t you say that before the syncline was imaged/interpreted 100 m away from where you see it?

L344: perhaps “low-cost” or “low-priced” or “economical” instead of “cheap”

L348: Quite unlikely that you don’t require any additional knowledge.

Author Response

We agree with all suggestions of the reviewers, and we have modified the manuscript in all clearly defined places. However, due to minor revision status and limited time for corrections, it was not possible to make significant rearrangements according to the second reviewer whose suggestions were unclear and vague, as he defined himself. Still, we tried to improve the manuscript to be clearer.

The most important changes in manuscript are:

More clear descriptions of some processing steps Added reference, as suggested by the reviewer Clarification of multiple sentences, according to the suggestions of reviewers Removed information about GPR due to low quality of data Some minor grammar corrections